# CONTEXT BASED MACHINE TRANSLATION WITH RECURRENT NEURAL NETWORK FOR ENGLISH-AMHARIC TRANSLATION

## ABSTRACT

The current approaches for machine translation usually require large set of parallel corpus in order to achieve fluency like in the case of neural machine translation (NMT), statistical machine translation (SMT) and example-based machine translation (EBMT). The context awareness of phrase-based machine translation (PBMT) approaches is also questionable. This research develops a system that translates English text to Amharic text using a combination of context based machine translation (CBMT) and a recurrent neural network machine translation (RNNMT). We built a bilingual dictionary for the CBMT system to use along with a large target corpus. The RNNMT model has then been provided with the output of the CBMT and a parallel corpus for training. Our combinational approach on English-Amharic language pair yields a performance improvement over the simple neural machine translation (NMT).

## 1 INTRODUCTION

Context based machine translation (CBMT) is a phrase-based machine translation (PBMT) approach proposed by Miller et al. (2006). Unlike most PBMT approaches that rely on statistical occurrence of the phrases, CBMT works on the contextual occurrence of the phrases. CBMT uses bilingual dictionary as its main translator and produces phrases to be flooded into a large target corpus.

The CBMT approach addresses the problem of parallel corpus scarcity between language pairs. The parallel corpus set for English-Amharic language pair, for instance, composes of the Bible, the Ethiopian constitution and international documents. These sources use words specific to their domain and overlook phrases and words used by novels, news and similar literary documents. The CBMT uses synonyms of words in place of rare words and rely on large target corpus and a bilingual dictionary to help with data scarcity(Miller et al., 2006). It is not dependent on large parallel corpus like most PBMT such as the statistical machine translation (SMT)(Brown et al., 1990) and the example-based machine translation EBMT(Gangadharaiah, 2011). The CBMT, however, fails in fluently translating texts compared to the neural machine translation (NMT).

The NMT learns the pattern of humans' translation using human translated parallel corpus. Its translations are more fluent and accurate than all the rest so far when evaluated individually (Popovic, 2017). However, NMT struggles to translate properly rare words and words not commonly used(Wu et al., 2016). In addition, NMT requires large parallel corpus for training.

The aim of this research is to build a system by combining the CBMT with the NMT for English to Amharic translation. The combination of PBMT and NMT is the future and most promising than the individual approaches themselves (Popovic, 2017). CBMT's ability to address rare words and the NMT's ability to produce fluent translation along with their context awareness makes them complementary couple.

The combination is done by providing the NMT with two inputs, one from the source language and the other from the output of the CBMT to produces the final target sentence. In this paper, we show that this approach utilizes the strength of each method to achieve a significant translation performance improvement over simple NMT. The improvement is mostly dependent on the performance of the CBMT and mostly on the bilingual dictionary of the CBMT.

## 2 RELATED WORKS

PBMT approaches are mostly used to translate English to Amharic as in the case of Gasser (2012),Tadesse & Mekuria (2000), Teshome (2000), Besacier et al. (2000), Zewgneh (2017) and Taye et al. (2015) . Below we summarize the researches with most significance to ours.

The SMT approach takes a parallel corpus as an input and it selects the most frequent target phrase based on statistical analysis for each searched source phrase (Brown et al., 1990). The SMT approach applied to the English-Amharic pair has produced 18.74 % BLEU score (Tadesse & Mekuria, 2000). The SMT has good accuracy in translating all the words in a sentence but it is not fluent (Oladosu et al., 2016).

Hybrid of SMT and rule based machine translation (RBMT) translates and orders the source text based on the grammar rules of the target language and sends it to the SMT for final translation(Yulianti et al., 2011)(Labaka et al., 2014). The hybrid approach for English-Amharic pair has achieved a 15% improvement over SMT on simple sentence and 20% improvement for complex sentence(Zewgneh, 2017). Hybrid of RBMT and SMT gets fluency from RBMT and accuracy from SMT but for longer sentences, the reordering fails(Oladosu et al., 2016).

The CBMT approach has been implemented for the language pair Spanish-English. In CBMT, the source phrases are translated using bilingual dictionary and flooded to target corpus. It has achieved 64.62% BLEU score for the researchers' dataset(Miller et al., 2006). The CBMT outperforms SMT in accuracy and fluency but translation of phrases with words not in the bilingual dictionary is weak (Miller et al., 2006).

The NMT has been researched by different groups and here the research by Googles' researchers on the language pair English-French is presented. The NMT model is trained using parallel corpus. The source sentence is encoded as a vector and then decoded with the help of an attention model. Googles' NMT model has achieved 38.95% BLEU score(Wu et al., 2016). The NMT has accuracy and fluency but it fails to translate the whole sentence and also fails to perform well with rare words(Wu et al., 2016). To solve this using sub-word units has been suggested (Sennrich et al., 2016) but Amharic has unique treats like "Tebko-lalto", one word with two different meanings, which can only be addressed by using context.

The NMT has been modified to translate low resourced languages. One approach uses universal lexical representation (ULR) were a word is represented using universal word embeddings. This benefits low resource languages which have semantic similarity with high resourced languages (Gu et al., 2018). This achieved 5% BLEU score improvement over normal NMT. However, most southern Semitic languages like Amharic do not have a strong semantic relative with large resource. NMT has also been modified to work with monolingual corpus instead of parallel corpus using cross-lingual word embedding(Artetxe et al., 2017). Such an approach achieved a 15.56% BLEU score which was less that the semi-supervised and supervised which achieved 21.81% BLEU score and 20.48% BLEU score respectively.

Combination of NMT and PBMT which takes the output of SMT (a PBMT) and the source sentence to train the NMT model has been used for the language pair English-German. It has achieved 2 BLEU points over basic NMT and PBMT(Niehues et al., 2016). Combination of NMT and PBMT which takes three inputs; output of basic NMT, output of SMT and output of Hierarchical PBMT (HPBMT) of SMT has been implemented for English-Chinese language pair. It achieved 6 BLEU points over basic NMT and 5.3 BLEU points over HPBMT(Zhang et al., 2017). The combination of PBMT and NMT performs better (Popovic, 2017) in terms of accuracy and fluency but it is dependent on the performance of the chosen PBMT approach.

## 3 METHODOLOGY

In this research, we have selected CBMT and NMT to form a combinational system. This approach addresses the limitation of context unawareness of some PBMT approaches like SMT and the need for large parallel corpus of simple NMT. In our approach, the source sentence in English and the translation output of the CBMT in Amharic has been fed to the NMT's encoder-decoder model as shown in Figure 1. The NMT model then produces the final Amharic translation.

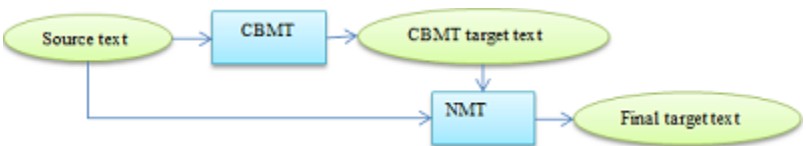

Figure 1: The proposed combination method

The combination of the CBMT and the NMT follows the mixed approach proposed by Niehues et al. (2016). Their mixed approach feeds the NMT with the source sentence and the output of the PBMT. The research by Zhang et al. (2017) also supports this way of combining different systems.

### 3.1 CBMT SYSTEM

The CBMT outperforms RBMT, SMT and EBMT when it comes to languages with less parallel corpora(Miller et al., 2006). It uses a bilingual dictionary, a large target corpus and smaller source corpus, which is optional. In the context based machine translation, there are different components working together to produce the translation. Figure 2 shows the flow of data in the different components of the CBMT.

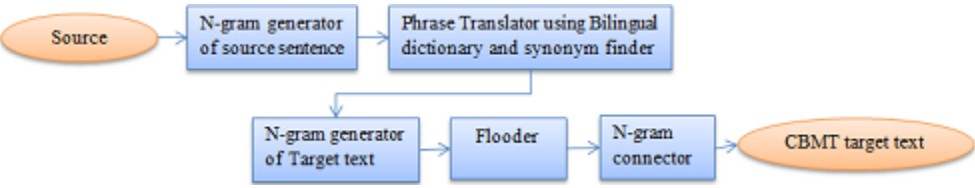

Figure 2: Overview of the CBMT system

The source sentence is converted into N-gram phrases and then it is translated using bilingual dictionary. CBMT's performance is mostly dependent on the efficiency of the dictionary. We have manually built a phrase based dictionary aided by Google translate. A synonym finder helps the dictionary's search using WordNet(Soergel, 1998). WordNet is a library with large lexical database of English words. It provides synonyms of the English words whose Amharic translations are not in dictionary. In this paper, a maximum of four N-grams has been used. Most phrases of English that are translated to a single word in Amharic have a length of four or less words.

For example the English phrase **"everyone who calls on the name of the lord will be saved"** has the translations in Output 1 using our dictionary.

```
[[['everyone', 'ሁላም ሰው'], ['everyone', 'ሁሉ'], ['everyone', 'ለሁላም'],
'everyone', 'ማንኛውም ሰው'], ['everyone', 'ለአይጋኖዳፉ']],
['who', 'ማን']],
['calls', 'ጥሪዎች'], ['calls', 'የሚጠራ'], ['calls', 'ከንደው']],
['on', 'ላይ']],
['name', 'ስም'], ['name', 'ስሙ'], ['name', 'ስሜም'], ['name', 'ለስሙህም']],
['of', ' r']],
['Lord', 'ጌታ'], ['Lord', 'የጌታን'], ['Lord', 'ጌታችን'], ['Lord', 'ከጌታም'],
'Lord', 'በጌታችን'], ['Lord', 'ጌታን'], ['Lord', 'ለጌታው'], ['Lord', 'ለጌታ'],
'Lord', 'የጌታ'], ['Lord', 'በጌታ'], ['Lord', 'የጌታቸንን'], ['Lord', 'ጌታችንን'],
'Lord', 'የጌታችን']],
['will', 'ፈቃድ']],
['will be saved', 'ይድናል']],
['be', 'መሆን']],
['saved', 'ይድናል'], ['saved', 'አንደንም'], ['saved', 'መዳናችን'], ['saved','ድናል'],
'saved', 'ይድና'], ['saved', 'አንዲድኑ'], ['saved', 'ትድናለህ'], ['saved','የምትድናውም'],
'saved', 'ይድኑ']]]
```

Output 1: The translated output of the N-grams

These translations have been combined into sentences in the same order as the source sentence. Then each sentence is converted into N-grams of variable length. The maximum flooded N-grams length is $\frac{len(Ngram)}{2} + 1$ if $len(Ngram) \geq 4$ else it is equal to *len(Ngram)*. This provides a good variable range to capture neighboring words in a single N-gram.

Output 2 shows sentences formed using the translations in Output 1. The N-grams to be flooded are formed by sliding one word from left to right through the combined sentence.

['ሁሉም ሰው ፕሬዎች ስም ጊታ ይድናል','ሁሉም ሰው ፕሬዎች ስም የጌታን ይድናል','ሁሉም ሰው የሚጠራ ስም ጊታ ይድናል',
'ሁሉም ሰው የሚጠራ ስም የጌታን ይድናል','ሁሉ ፕሬዎች ስም ጊታ ይድናል','ሁሉ ፕሬዎች ስም የጌታን ይድናል',
'ሁሉ የሚጠራ ስም ጊታ ይድናል','ሁሉ የሚጠራ ስም የጌታን ይድናል']

Output 2: The translated output combined into sentence

Output 3 shows the N-grams for the translated sentences shown in Output 2.

[['ሁሉም ሰው ፕሬዎች ስም', 'ሁሉም ሰው የሚጠራ ስም', 'ሁሉ ፕሬዎች ስም', 'ሁሉ የሚጠራ ስም'],
['ሰው ፕሬዎች ስም ጊታ', 'ሰው ፕሬዎች ስም የጌታን', 'ሰው የሚጠራ ስም ጊታ', 'ሰው የሚጠራ ስም
የጌታን', 'ፕሬዎች ስም ጊታ', 'ፕሬዎች ስም የጌታን', 'የሚጠራ ስም ጊታ', 'የሚጠራ ስም የጌታን'],
['ፕሬዎች ስም ጊታ ይድናል', 'ፕሬዎች ስም የጌታን ይድናል', 'የሚጠራ ስም ጊታ ይድናል', 'የሚጠራ ስም የጌታን ይድናል',
'ስም ጊታ ይድናል', 'ስም የጌታን ይድናል']]

Output 3: The N-grams for the translated sentences

The flooder is then responsible for searching the translated phrases in the target corpus and finding the longest N-gram match. For each phrase to be flooded, it selects a phrase in the target corpus with the most translated words and least in-between words amongst the words matched. The flooder produces the result in Output 4 with the Book of Romans as the target corpus to be flooded.

[[[['ሰው', 'በኩል', 'ወደ', 'ገሰፀ', 'አጋ', 'ዋ', 'ሁሉ', 'ሞተዋ', 'በንጣት', 'በኩል', 'ንባፈልል', 'ብሎ', 'መገጋ',
'ሞተ', 'ወደ', 'ስፈች', 'ሁሉ', 'መጣ', 'ያስንደዘም', 'ሁሉም'],['ስም', 'የሚጠራ'], ['ስም', 'የሚጠራ', 'ሁሉ']],
[['ጊታ', 'ስፈ', 'መገፅስትርርስት', 'ሰው'], ['የጌታን', 'ስም'], ['ስም', 'የሚጠራ'], ['የጌታን','ስም', 'የሚጠራ']],
[['ስም', 'የሚጠራ', 'ሁሉ', 'ይድናል'], ['የጌታን', 'ስም', 'የሚጠራ', 'ሁሉ', 'ይድናል']]]}

Output 4: Final output of flooder for single flooded file

The N-gram connector combines the flooded text to find the longest overlap of the translated target text. The overlapping system favors those with the least number of not searched words found in between the searched N-grams when calculating the overlap. Output 5 shows the final outcome of the N-gram connector.

The system selects the maximum or longest overlapping phrases from the combiner and merges them to form the final target sentence. So finally, the translation for the example English phrase **"everyone who calls on the name of the lord will be saved"** is 'የጌታን ስም የሚጠራ ሁሉ ይድናል'.

## 3.2 NMT SYSTEM

In this paper, we have used RNN (recurrent neural networks) for the NMT. In RNN, the output is fed back to the neuron to learn from both the fresh input and its previous output. This improves RNN's performance because it learns from its errors while training.

The neural cell used is the LSTM (long short-term memory), introduced by Hochreiter & Schmidhuber (1997). We have used LSTM cells for both the encoding and decoding of the sentences. For the decoding, a greedy algorithm has been used. The algorithm selects the first fit word that has the highest probability of occurrence. Probability refers to the probability of being translated and appearing next to the word before itself.

The system has an attention layer between the encoder layer and the decoder layer. We have used the Luong attention model(Luong et al., 2015). Equation 1 through Equation 4 shows Luong attention model's computation.

$$\alpha_{ts} = \frac{exp(score(h_t, \overline{h}_s))}{\sum_{s'=1}^{S} exp(score(h_t, \overline{h}_{s'}))} \quad \textbf{[Attention Weights]} \quad (1)$$

[[['ንም', 'የኔ... '], ['ንም', 'የኔ... ']],[['ፍጻጉ', 'ንም', 'የኔ... '], ['ፍጻጉ', 'ንም','የኔ... ', 'ውሉ', 'ጸዩ...']],[['ፍጻጉ', 'ንም', 'የኔ... ', 'ውሉ', 'ጸዩ... '], ['ፍጻጉ', 'ንም', 'የኔ... ', 'ውሉ','ጸዩ...']]]

Output 5: Final Output of N-gram connector

$$c_t = \sum_s \alpha_{ts} \overline{h}_s \qquad \textbf{[Context vector]} \qquad (2)$$

$$a_t = f(c_t, h_t) = tanh(W_c[c_t : h_t]) \quad \textbf{[Attention Vector]} \qquad (3)$$

$$score(h_t, \overline{h}_s) = h_t^T W \overline{h}_s \quad \textbf{[Luong's multiplicative style]} \qquad (4)$$

The score function, calculated using Equation 4, is used to compare the output of the decoder $(h_t)$ with the output of the encoder $(h_s)$ in order to find the attention weight calculated using Equation 1. The Attention weights $(alpha_{ts})$ are then used for the context vector$(c_t)$ calculated by Equation 2. This context vector as well as the output of the decoder is then used to produce the final output of the decoder using Equation 3.

### 3.3 COMBINATION OF CBMT AND NMT

To combine the two systems, we have made the NMT model to accept two inputs. We have used the proposed method of combining PBMT with NMT accepting two source inputs by Zoph & Knight (2016). According to Zoph & Knight (2016), having two inputs, where one is the source sentence and the other a translation of the source to another language different from the target, helps the NMT produce a better result.

The source sentence and the sentence translated using the CBMT are encoded separately and are given to the attention layer. The attention layer focuses on the two inputs at the same time rather than separately. There is a single decoder, which receives the output of the attention layer and provides the final translation.

Based on the paper Zoph & Knight (2016), the final outputs of the encoders $(\overline{h}_t)$ are concatenated and a linear transformation is applied to the concatenated output which is activated by *tanh* using Equation 5. On the other hand, the final states $(c_t)$ are simply added as shown by Equation 6. In the attention, the different context vectors are calculated separately and concatenated to produce the final output of the attention vector based on the Luong attention mechanism(Luong et al., 2015) using Equation 7.

$$h = tanh(W_c[h_1; h2]) \qquad (5)$$

$$c = c_1 + c_2 \qquad (6)$$

$$\overline{h}_t = tanh(w_c[h_t; c_t^1; c_t^2] \qquad (7)$$

## 4 EXPERIMENT SETUP

We evaluate the proposed approach for the language pair English-Amharic using the same training parameters for both the basic NMT and the combinational system. The encoder-decoder setup has 1024 LSTM cells or hidden units with 1024 word embedding and the data has been trained for 100 epochs.

### 4.1 CORPUS USED

This research uses the new International Version Bible because both the Amharic and English versions are translated from the same Dead Sea scrolls. This makes it more accurately parallel than other versions of the Bible translation.

The whole New Testament of the Bible has been used as a corpus providing a total of 8603 phrases and sentences. We have used two books of the Bible, Paul's letter to the Romans (Romans) and the Gospel according to Mark (Mark), as a test set.

Google translate has been used as the main agent of translation for the manually built bilingual dictionary used by the CBMT. 77% of the total 6,793 vocabulary words have been translated using

Google. In addition to Google translate; we have done a manual translation for 1,426 vocabulary words in the book of Romans and other 150 vocabulary words using the Bible.

Manual translation here refers to the translation of each word using every entry it has in the Bible by a human. Figure 5 shows the outcome of such translation for the word *Acknowledge*. Manual translation helps to address the variations in Amharic translated words caused by gender (female or male), plural form and the different persons (first, second and third persons). English words do also have different Amharic translations based on their context as shown in Figure 3. *Acknowledge* has been translated into four main stem words እወቀ, አስበ, ተቀበለ and መስከረ

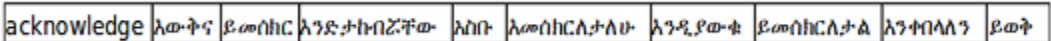

Figure 3: Manual translation result of the word acknowledge

### 4.1.1 DATASET

We have fed the same dataset to all systems with minor variations. In the CBMT, we have used the book of Mark and the book of Romans as a test set. The flooded texts for the book of Romans were the book of Romans itself and Paulian epistles without Romans. The flooded texts for the book of Mark were the book of Mark itself and the gospels without Mark. The books have been flooded to themselves in order to evaluate the performance of the CBMT when the searched text is found in the flooded text and also to see the impact of the bilingual dictionary on the CBMT.

The combinational system has two different test sets and different models. The first test set has the output of the CBMT and the source sentence as an input for the NMT. The second test set gives the original target text and the source sentence as an input to the NMT models and we have called it the ideal approach. This was done so to see the impact of the CBMT output and the errors introduced by the CBMT on the combinational system.

In the basic NMT and combinational system, similar dataset as the CBMT is used. We have used Paulian epistle without Romans to train a basic NMT model and the combinational model. Then we have tested the model using the book of Romans as the test or holdout set. We have used 10-fold cross validation(Dietterich, 1998) to train and test the basic NMT model and the combinational model with Romans. In a similar manner, we have used 10-fold cross validation(Dietterich, 1998) to train the basic NMT model and the combinational model with the book of Mark. We have also used holdout validation(Raschka, 2018) with a random 80% training and 20% test data split alongside the 10-fold cross validation for both Mark and Romans to obtain a more general representation of the results.

## 5 EVALUATION METHOD

We have measured the translation performance based on the fullness of the translation, whether the system translates all words; context awareness, whether the translation is true to the context and Coherence, whether the translated sentence has a smooth flow of words in terms of syntax.

In this research, the BLEU score is the chosen method of evaluation. It answers all the above-enlisted criteria. The quality of translation is measured by the correspondence between a machine's output and that of a human(Kishore et al., 2002). BLEU score is defined in the range between 0 and 1 (or in percentage between 0 and 100) where 1 is a perfect match with the reference and 0 is for no words matched.

## 6 RESULTS AND DISCUSSION

This section provides the results obtained along with a brief explanation of the factors and the components.

## 6.1 CBMT Result and Discussion

The system has been tested using a custom-made dictionary using Google translate and manual translation. We have generated the vocabulary of the dictionary from the English version of the NIV Bible. Table 1 depicts the CBMT test results obtained, using BLEU score evaluation method. We

Table 1: Results for the CBMT

|   | Flooded Data | Test data | BLEU score |
|---|---|---|---|
| 1 | Paulian epistle without Romans | Romans | 27.91 |
| 2 | Romans | Romans | 70.46 |
| 3 | Mark | Mark | 21.98 |
| 4 | Gospels without Mark | Mark | 21.43 |

have implemented manual translation for the book of Romans on about 80% of its total vocabulary. Hence, it has a better performance yield than the book of Mark, whose translation was solely dependent on Google translate. This is so both when they are flooded to the text that contained them (by 48%) and when they are flooded to the text without them (by 6%). However, the translation of Romans does not produce a 100% as would be expected when it is part of the flooded document. This is mainly because the system selects the overlapping N-gram based on the number of words matched, two consecutive phrases that may have a high overlap but which are not the correct ones may be selected.

## 6.2 NMT Result and Discussion

The NMT test results obtained using BLEU score evaluation method are depicted in Table 2. The

Table 2: Results for the NMT

|   | Training input Data | Test data | Validation | BLEU score |
|---|---|---|---|---|
| 1 | Paulian epistle without Romans | Romans | Holdout | 10.24 |
| 2 | Romans | Romans | 10-fold | 11.95 |
| 3 | Mark | Mark | 10-fold | 12.42 |
| 5 | Romans | Romans | Holdout | 7.28 |
| 6 | Mark | Mark | Holdout | 10.12 |

test result obtained from Mark was better than that from Romans by an average of 1.62%. Although insignificant difference, it attributes to Marks' writing having similar words unlike the diverse word selection in Romans(Clay, 2018).

## 6.3 CBMT and NMT Combination Result and Discussion

There are two test cases for this section. In the first case, we have given the NMT the source sentence and the output of the CBMT as an input per the proposed methodology. Table 3 shows the results obtained from such a setup. In the second case, we have given the NMT the English source sentence

Table 3: Results for the combination of NMT and CBMT

|   | Training input Data | Test data | Validation | BLEU score |
|---|---|---|---|---|
| 1 | Paulian epistle without Romans | CBMT output Romans | Holdout | 11.55 |
| 2 | Romans | CBMT output Romans | 10-fold | 14.07 |
| 3 | Mark | CBMT output Mark | 10-fold | 12.36 |
| 5 | Romans | CBMT output Romans | Holdout | 13.84 |
| 6 | Mark | CBMT output Mark | Holdout | 12.73 |

and the original Amharic as an input creating an ideal system. The test results obtained using BLEU score evaluation method, are depicted in Table 4 for the ideal combinational system. In the first case, when the CBMT output is used as an input to the NMT, the Book of Romans performed better than

Table 4: Ideal case Results for the combination of NMT and CBMT

|   | Training input Data | Test data | Validation | BLEU score |
|---|---|---|---|---|
| 1 | Paulian epistle without Romans | original Amharic Romans | Holdout | 11.63 |
| 2 | Romans | original Amharic Romans | 10-fold | 18.46 |
| 3 | Mark | original Amharic Mark | 10-fold | 17.41 |
| 5 | Romans | original Amharic Romans | Holdout | 25.74 |
| 6 | Mark | original Amharic Mark | Holdout | 25.52 |

the book of Mark by 1.71%. The CBMT output of Romans is better than that by the book of Mark and its impact has propagated to the combinational system. In the ideal case scenario the results is more or less the same. The result for the book of Romans was better than the book of Mark by only 0.63%.

## 6.4 DISCUSSION OF ALL RESULTS

The average of the results obtained from the systems have been calculated and shown in Tables 5 for comparison. The ideal combinational system, which takes the original target Amharic text as the

Table 5: Summary of all results

| System | Dataset | BLEU score |
|---|---|---|
| CBMT | book of Romans | **27.91** |
| | book of Mark | 21.43 |
| simple NMT | book of Romans | 9.61 |
| | book of Mark | 11.27 |
| Combinational system with CBMT output | book of Romans | 13.95 |
| | book of Mark | 12.54 |
| Combinational system with original text (ideal) | book of Romans | 22.1 |
| | book of Mark | **21.46** |

second input, has performed better on average with 11.34 BLEU score gain over the NMT. The ideal system, however, did not outperform the CBMT on average but produced results in the same range. The combinational system with CBMT output given as the second input for the NMT, achieves 2.805 BLEU score point over the simple NMT. The CBMT without being provided the target flooded data has performed better by 14.23 BLEU points over the simple NMT.

## 7 CONCLUSION

The research set out to find a combinational system with components that complement each other when translating a document from English to Amharic. We have proposed the CBMT and the NMT as the complementing pair in terms of parallel data size, accuracy, context awareness with translation and coherence.

The CBMT system performed better than the basic NMT and the combinational system given the same size of data. However, the ideal combination of the CBMT and NMT has a BLEU score in the range of the CBMT while outperforming the simple NMT by 11.34 BLEU points. For the book of Mark whose writing resembles the other Gospels, the ideal combination outperformed all. This entails that with smaller increase in parallel corpus for the NMT of the ideal system will outperform both individual systems. The output from the CBMT has a great impact on the performance of the combinational systems as seen by the performance of the proposed combinational system compared to the ideal system. The proposed combinational system still has outperformed the basic NMT by 2.805 BLEU score points in spite of the errors introduced by the CBMT.

Therefore, a CBMT with a well-built bilingual dictionary that produces a close to ideal output along with a well-trained NMT with sufficient data makes a fluent combinational system that outperforms a simple NMT and a basic CBMT system. In conclusion, the study suggests the combinational system for translation of English to Amharic language.

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
