# OpenReview forum: "Context Based Machine Translation With Recurrent Neural Network For English-Amharic Translation "
_ICLR.cc/2020/Conference — Reject_

### Official Review · AnonReviewer2 · 2019-10-23
**Official Blind Review #2**

**Rating:** 1

**Review:**

For the low-resource pair English-Amharic, the authors propose to combine context-based machine translation (CNMT), which is built by using a bilingual dictionary and then connecting the resulting n-grams, with a neural MT system. The source sentence, as well as the CNMT output, are used as inputs to the RNN-based translation system.

I vote for rejection because the paper makes some unfounded claims, misses important related work, has some methodological issues and presents unconvincing results.

The paper claims that CBMT is a new approach, but it dates from 2006. The authors say that it outperforms other MT approaches, but a more recent reference would be needed. While neural machine translation may sometimes struggle with rare words, using sub-word units may help alleviate the issue (Sennrich et al. Neural machine translation of rare words with subword units). The claim that RNNs learn from their previous mistakes is also unclear. It's true in the sense that backpropagation is learning from errors, but using previous reference outputs can cause exposure bias (Ranzato et al. Sequence Level Training with Recurrent Neural Networks).

The paper fails to cite related work, in particular on low-resource NMT (e.g. Gu et al. Universal Neural Machine Translation for Extremely Low Resource Languages) and unsupervised translation (e.g. Artetxe et al. Unsupervised Neural Machine Translation).

The CBMT system is built on top of the Google Translate English-Amharic system. However, that model may have seen the test data during training.

By combining CBMT with NMT, the authors obtain better results than NMT alone, but worse than with CBMT only. As such, the usefulness of the approach in a very low-resource scenario is unclear.

Minor points:

Some typos could be corrected: BLUE -> BLEU, Loung -> Luong, weather -> whether

**Experience Assessment:**

I have published in this field for several years.

**Review Assessment: Checking Correctness Of Derivations And Theory:**

N/A

**Review Assessment: Checking Correctness Of Experiments:**

I assessed the sensibility of the experiments.

**Review Assessment: Thoroughness In Paper Reading:**

I read the paper at least twice and used my best judgement in assessing the paper.

---

> ### Author Response · Authors · 2019-11-14
> **Thank you;  CBMT is new compared to other PBMT approaches, sub-word unit may not be ideal for Amharic,  and our final claim holds in the context of having a strong bilingual dictionary and an improved NMT with larger corpus as mentioned in the paragraph above it.**
>
> Thank you for taking the time to read our paper twice and giving us your comments.
>
> Re: “The paper claims that CBMT is a new approach, but it dates from 2006.”
>
>  When we said it is new, we were strictly comparing it to other Phrase Based Machine Translation(PBMT) approaches such as the Statistical Machine Translation(SMT), Rule-Based Machine Translation (RBMT) and Example-based Machine Translation(EBMT). In addition, the original paper on CBMT is the one claiming its better performance over other PBMT such as the SMT and the RBMT (Miller et al. Context-based machine Translation). These, however, are good points we should clarify in the paper.
>
> Re: using sub-word units may help alleviate the issue.
>
> The paper mentioned (Sennrich et al. Neural machine translation of rare words with subword units) does have its own merit. However, the three focus areas (names (via character copying or transliteration), compounds (via compositional translation), and cognates and loanwords (via phonological and morphological transformations)) do not cover most translation problems in Amharic. For example: there is such a thing in Amharic called ጠብቆ-ላልቶ “tebko-lalito”; in which the same word has two different meanings based on the pronunciation of a syllable in the word. Example: “ያነባሉ” “yanebalu” refers to reading when read stressing the ‘ba’ (ጠብቆ-tebko) on the other hand when it is read without stressing on the ‘ba’ (ላልቶ-lalito) it means to cry. Such unique treats of Amharic can be addressed by seeing the context than any other methods proposed by the paper. Besides Tebko-lalto, Amharic has different meanings for the same word based on the context of the text. Our paper gives an example for this; the word ‘acknowledge’ could mean ‘to know’, ‘to think’, ‘to accept’ or ‘to testify’ based on the context of the sentence.
>
> Re:  The paper fails to cite related work, in particular on low-resource NMT (e.g. Gu et al. Universal Neural Machine Translation for Extremely Low Resource Languages) and unsupervised translation (e.g. Artetxe et al. Unsupervised Neural Machine Translation)
>
> These surely are important papers to have cited. Thank you for bringing them to our attention, we will include them in the literature/related works section. The first paper by the Google researchers is universal and could be used for Amharic. However, it relies on having highly resourced semantically related language to help the low resourced languages. Amharic being a Semitic language, the closest highly resourced language to it would be Hebrew. However, Amharic has been highly influenced grammatically and semantically by Cushitic languages such as Agew and as a southern Semitic language, it is quite different from the northern Semitic languages like Hebrew and Arabic.
> The unsupervised translation paper on the other hand uses a monolingual corpus which makes it ideal. The result shows the unsupervised is not as effective as semi-supervised and supervised. Although small, we do have a parallel corpus for the English-Amharic pair; hence the unsupervised does not seem to be ideal for the pair.
>
> Re: “The CBMT system is built on top of the Google Translate English-Amharic system. However, that model may have seen the test data during training.”
>
> We fear there might be a misunderstanding; the reviewer seems to may have believed that the CBMT is a model to train and using Google translate makes the system biased. However, Google translate is used in our work as a dictionary and nothing else.
>
> Re:”By combining CBMT with NMT, the authors obtain better results than NMT alone, but worse than with CBMT only. As such, the usefulness of the approach in a very low-resource scenario is unclear.”
>
>  We have made that claim on the bases of having a well-built bilingual dictionary producing a close to ideal CBMT output and combined with the NMT with increased size of corpus. This claim comes from the results for the book of Marks; the combinational system with the ideal setup (less errors introduced by the CBMT) performed better than all other systems (0.03 BLEU score over CBMT). It is known for neural network, size matters as stated by the paper (D. Ellis and N. Morgan. Size matters: an empirical study of neural network training for large vocabulary continuous speech recognition). However, we did not have large data available for us to work on; still from our perspective; the small data gives out a sufficient premise to make the claim within the conditions of providing larger data and well built CBMT.

---

### Official Review · AnonReviewer3 · 2019-10-25
**Official Blind Review #3**

**Rating:** 1

**Review:**

This paper aims to combine a traditional CBMT system with an NMT system. The core idea of the paper is to use the output of the CBMT system as a second source to a multi-source NMT system. The first source of the system is CBMT, the second source is the original source and the output is the translation in the target language. All the experiments are conducted for English-Amharic with a small amount of parallel data from the Bible. A lot of details are provided about the generation of the CBMT system using Google Translate and details are provided about the approach to create such a system.

Pros:
- The idea of using additional outputs to the NMT system and outputs from a context-aware system is neat. This has been done by others who have included PBMT outputs in the past. However, this might be the first to include a CBMT result as an additional source.
- Focusing on a low-resource language like Amharic is good for the community and will encourage more research in these underrepresented languages.

Cons:
- A lot of the techniques described for building the traditional CBMT system are obsolete these days and people prefer neural methods. I worry if this is relevant in the current day.
- The authors could have compared against other ways of incorporating context as a strong baseline - like a n-to-n or n-to-1 NMT system.
- Most experiments in the paper are conducted on a small data set and this is a big downside of the paper.
- More detailed analyses of where the contextual phenomena was incorporated might have helped the paper.



**Experience Assessment:**

I have published one or two papers in this area.

**Review Assessment: Checking Correctness Of Derivations And Theory:**

I carefully checked the derivations and theory.

**Review Assessment: Checking Correctness Of Experiments:**

I carefully checked the experiments.

**Review Assessment: Thoroughness In Paper Reading:**

I read the paper thoroughly.

---

> ### Author Response · Authors · 2019-11-14
> **Thank you; CBMT is not obselete yet and please do consider that we are working with the small data readily available to us.**
>
> We are thankful for the carefully listed comments and find them both (pros and cons) helpful.
> Re:  “A lot of the techniques described for building the traditional CBMT system are obsolete these days and people prefer neural methods. I worry if this is relevant in the current day.”
>
>  Combining or hybridization of Phrase Based Machine Translation (PBMT) and NMT is still being researched as stated by one of the papers cited (Popovic, 2017. Comparing Language Related Issues for NMT and PBMT between German and English). CBMT, a recent approach among the PBMT approaches, does still stand to contend when combined with a neural method.
>
> Re:  Most experiments in the paper are conducted on a small data set and this is a big downside of the paper.
>
> You do have a good point and we do agree with you; but please do consider that we are working with the small data readily available to us with the limited computational resource. We have also indicated that the Amharic language is a low resourced language.

---

### Official Review · AnonReviewer1 · 2019-10-27
**Official Blind Review #1**

**Rating:** 1

**Review:**

This paper presents a machine translation system based on a combination of a neural machine translation system (NMT) and a context-based machine translation (CBMT). The method is evaluated on a small parallel corpus application of English-Amharic translation. The idea is that in the small corpus setting, the CBMT can leverage a manually built bilingual dictionary
to improve on the standard NMT.

Clarity: The paper is reasonably clear, though there are numerous typos and minor language glitches (missing articles, etc.). Overall the quality of the writing is probably a bit below what's acceptable for publication at ICLR, but nothing that could not be fixed on subsequent revisions.

Novelty: The method proposed appears to be a fairly straightforward variant of one proposed in a previous paper, where an NMT system was combined with a phrase-based MT system (Zoph & Knight, 2016). There seems to be no novel machine learning contribution (nor is it claimed). This paper seems more appropriate for a venue more focused on machine translation rather than a machine learning venue such as ICLR.

Empirical evidence in support of the claims: The authors set out to demonstrate that by combining a CBMT output into an NMT approach, one can get the best of both approaches. Their results do not strongly support this claim. The results suggest that in the context of the small-scale experiments considered, the baseline CBMT model is actually overall the best performing model. It is therefore strange that, in their last sentence of the conclusion, the authors persist in claiming that their combination "outperforms a simple NMT and a basic CBMT system". That being said, the sub-claim that the
NMT/CBMT hybrid improves on the baseline NMT system is well established.

In light of the relatively low novelty and the lack of compelling empirical performance for the proposed combined MT system, I do not feel that this paper is appropriate for ICLR at this time.


**Experience Assessment:**

I have published one or two papers in this area.

**Review Assessment: Checking Correctness Of Derivations And Theory:**

I carefully checked the derivations and theory.

**Review Assessment: Checking Correctness Of Experiments:**

I carefully checked the experiments.

**Review Assessment: Thoroughness In Paper Reading:**

I read the paper thoroughly.

---

> ### Author Response · Authors · 2019-11-14
> **Thank you; But does not the ICLR accept application of Machine Learning in Natural Language Processing? For the Epirical evidence comment; the claim holds in the context of having a strong bilingual dictionary and an improved NMT with larger corpus as mentioned in the paragraph above it.**
>
> Thank you for your valuable feedback. We shall correct the typos, fix the missing articles, and send the corrected version.
>
> We agree with you that we have not introduced a new approach to machine learning when combining the CBMT and the NMT. However, none other has attempted combining the CBMT and NMT for low resource languages. In addition, the ICLR page does state that it accepts papers related to “applications in vision, audio, speech, natural language processing, robotics, neuroscience, computational biology, or any other field”. This is an application of machine learning in natural language processing.
>
> Re: “It is therefore strange that, in their last sentence of the conclusion, the authors persist in claiming that their combination "outperforms a simple NMT and a basic CBMT system".
>
>  We have made that claim on the bases of having a well-built bilingual dictionary producing a close to ideal CBMT output and combined with the NMT with increased size of corpus. This claim comes from the results for the book of Marks; the combination system with the ideal setup (less errors introduced by the CBMT) performed better than all other systems (0.03 BLEU score over CBMT). It is known for neural network, size matters as stated by the paper (D. Ellis and N. Morgan. Size matters: an empirical study of neural network training for large vocabulary continuous speech recognition). However, we did not have large data available for us to work on; still from our perspective; the small data gives out a sufficient premise to make the claim within the conditions of providing larger data and well built CBMT.

---

> > ### Comment · AnonReviewer1 · 2019-11-15
> > **reply**
> >
> > Regarding the suitability for ICLR: I agree that machine translation is well within the domain of interest of ICLR participants. The issue is w.r.t. the low novelty and significance of this specific paper - in the context of a machine learning conference. For a more focused conference on machine translation - this work could be perfectly appropriate.

---

### Decision · Program_Chairs · 2019-12-19

**Decision:**

Reject

**Comment:**

The authors propose a model which combines a neural machine translation system and a context-based machine translation model, which combines some aspects of rule and example based MT.  This paper presents work based on obsolete techniques, has relatively low novelty, has problematic experimental design and lacks compelling performance improvements. The authors rebutted some of the reviewers claims, but did not convince them to change their scores.